# Patterns of Equine Small Strongyle Species Infection after Ivermectin Intervention in Thailand: Egg Reappearance Period and Nemabiome Metabarcoding Approach

**DOI:** 10.3390/ani14040574

**Published:** 2024-02-08

**Authors:** Mohamed H. Hamad, Sk Injamamul Islam, Wanarit Jitsamai, Teerapol Chinkangsadarn, Darm Naraporn, Suraseha Ouisuwan, Piyanan Taweethavonsawat

**Affiliations:** 1The International Graduate Program of Veterinary Science and Technology (VST), Faculty of Veterinary Science, Chulalongkorn University, Bangkok 10330, Thailand; dr.mohamedhossam92@gmail.com (M.H.H.); injamamulislam017@gmail.com (S.I.I.); 2Department of Animal Infectious Diseases, Faculty of Veterinary Medicine, Zagazig University, Zagazig 44511, Egypt; 3Parasitology Unit, Department of Veterinary Pathology, Faculty of Veterinary Science, Chulalongkorn University, Bangkok 10330, Thailand; 4Department of Parasitology and Entomology, Faculty of Public Health, Mahidol University, Bangkok 10400, Thailand; neen_c6h12o6@hotmail.com; 5Department of Surgery, Faculty of Veterinary Science, Chulalongkorn University, Bangkok 10330, Thailand; teerapol.c@chula.ac.th; 6Horse Farm and Laboratory Animal Breeding Center, Queen Saovabha Memorial Institute, The Thai Red Cross Society, Hua-Hin 77110, Thailand; darm_ae@windowslive.com (D.N.); suraseha@yahoo.com.sg (S.O.); 7Biomarkers in Animals Parasitology Research Unit, Chulalongkorn University, Bangkok 10330, Thailand

**Keywords:** ivermectin, egg reappearance period, strongyle, ITS-2 metabarcoding, horse

## Abstract

**Simple Summary:**

Anthelmintic resistance has become a global concern due to the ill-informed use of deworming medications to control parasitic infections in horses. In Thailand, ivermectin has been the primary drug for deworming horses over an extended period. However, there are no available data on the use of ivermectin for treating strongyle infections in domesticated horses in Thailand. The study objectives were to evaluate the performance of ivermectin in treating strongyle infections, employ high-throughput sequencing to explore variations in infection patterns among species, and identify the species responsible for early egg shedding. The findings demonstrated that ivermectin successfully eliminated adult strongyle parasites within two weeks. However, the period for egg reappearance unexpectedly shortened to 6 weeks post-treatment. This early egg shedding was primarily associated with certain species, indicating different reemerging patterns. The information gained is valuable for developing effective strategies to control strongyle infections in horses, contributing to the overall well-being of these animals and the sustainability of equine healthcare.

**Abstract:**

The indiscriminate use of anthelmintics to control parasitic nematodes in horses has led to the emergence of anthelmintic resistance worldwide. However, there are no data available on using ivermectin for treating strongyle infections within domesticated horses in Thailand. Therefore, this study aimed to use the fecal egg count reduction (FECR) test to determine the strongylid egg reappearance period (ERP). Additionally, the nemabiome metabarcoding approach is incorporated to study patterns of strongyle species infection following ivermectin treatment. The study results indicate that, although ivermectin effectively eliminated adult strongyle parasites within two weeks post-treatment, the ERP was shortened to 6 weeks post-treatment with a mean FECR of 70.4% (95% CI 46.1–84.0). This potentially indicates a recent change in drug performance. In addition, nemabiome metabarcoding revealed that strongyle species have different levels of susceptibility in response to anthelmintic drugs. The reduction in ERP was associated with the early reappearance of specific species, dominated by *Cylicostephanus longibursatus* and *Cylicocyclus nassatus*, indicating the lower susceptibility of these species. In contrast, *Poteriostomum imparidentatum*, *Triodontophorus nipponicus*, and *Triodontophorus serratus* were not found post-treatment, indicating the high level of susceptibility of these species. This information is vital for comprehending the factors contributing to the emergence of resistance and for devising strategies to manage and control strongyle infections in horses.

## 1. Introduction

Equine strongylid parasites are widely prevalent among grazing horses across the globe. Over 50 species of Strongylinae (large strongyles) and Cyathostominae (small strongyles) subfamilies have been identified [1,2]. Although large strongyles can have severe pathogenic effects, small strongyles are more commonly observed in equine populations [3,4,5]. The co-infection of an individual host with 15–25 cyathostomin species has become commonplace [6,7,8]. Moreover, the prevalence of cyathostomins is associated with a disease syndrome known as Larval Cyathostominosis, which is characterized by diarrhea, hypoproteinemia, and reduced growth rates [9,10].

Equine strongyle infections are controlled by three classes of anthelmintics. Benzimidazoles (BZs), such as fenbendazole (FBZ) [11,12], tetrahydropyrimidines (THPs), and pyrantel pamoate (PYR) [13,14], have been widely used for many decades [15,16]. Resistance against these classes has been documented in numerous reports, particularly with excessive and frequent administration [17,18,19]. As a result, deworming programs targeting strongyle infections rely upon macrocyclic lactone compounds (MLs). Among these, ivermectin (IVM) and moxidectin (MOX) are frequently and interchangeably used. However, the cost of MOX is relatively higher than that of IVM [20], and ongoing efforts seek to preserve the efficacy of MOX against encysted larvae by limiting its excessive usage [21,22]. Therefore, IVM has been established as the primary anthelmintic drug for the management of strongyle infections in current practice [23,24,25,26]. However, the widespread use of IVM has been accompanied by reports of diminished effectiveness and the emergence of resistance in cyathostomin populations around the world [19,27,28,29].

The Fecal Egg Count Reduction Test (FECRT) has historically held a central role in equine veterinary medicine for evaluating the efficacy of deworming treatments and detecting anthelmintic resistance [30]. Therefore, the World Association for the Advancement of Veterinary Parasitology (WAAVP) introduced a new guideline for FECRT that is firmly grounded in a robust statistical framework, offering a more sensitive assessment of the deworming treatment efficacy in livestock [31]. This approach employs two separate statistical tests to classify the anthelminthic drugs based on the FECRT result and recommends a 90% confidence interval (CI) to enhance the precision and rigor of this classification process [31].

The estimation of egg reappearance periods (ERPs) was proposed more than two decades ago as a potential method for detecting emergent resistance in cyathostomins [32]. However, recent investigations contradict this notion. As the interpretation of shortened ERPs lacks clarity, they cannot not be solely considered as conclusive evidence of emerging anthelmintic resistance [33]. Nonetheless, it is essential to acknowledge that this shift in drug performance represents a significant departure from historical patterns, carrying evident implications for controlling strongyle parasites [31]. Thus, the WAAVP has recently published a guideline for measuring egg reappearance periods (ERPs) in equine species [34]. Previous reports have shown that the ERP in IVM-treatment is typically 8–13 weeks [35,36,37]. However, recent studies have indicated a reduced ERP 6–8 weeks after treatment [38,39]. The rationale behind this phenomenon may be attributed to an increased selection pressure for resistance development in specific species or a potential reduction in the efficacy of anthelmintics against the luminal stages of cyathostomins [32].

Cyathostomins encompass a remarkably large number of species co-infecting equine hosts. However, limited information is currently available regarding species-specific ERP values, and the sensitivity of individual species to anthelmintic drugs, since it is unlikely that the 50 species would display a uniform resistance pattern [40,41]. A previous study identified four cyathostomins (*Cyathostomum catinatum*, *Cylicocyclus* (*Cyc.*) *nassatus*, *Cyc. radiatus*, and *Cylicostephanus longibursatus*) associated with the early reappearance of eggs following a 4-week treatment with IVM [42]. Additionally, another study found that the majority of luminal fourth-stage larvae (L4s) that remained viable two weeks after IVM treatment were morphologically identified as *Cyc. insigne* [43]. These findings emphasize the need for a greater investigation into cyathostomin communities that are responsible for developing resistance. The identification of cyathostomin species based on their morphological characteristics requires a high level of expertise and experience [27], and thus may explain the limited number of studies undertaken in this area. Transcribed spacer-2 (ITS-2) rDNA nemabiome metabarcoding has emerged as a potential non-invasive method to characterize the mixed strongyle infections in horses [44]. This methodology not only enables the precise and reproducible quantification of species diversity within parasite communities but also has the potential to identify low presented species in given samples [42,45,46]. Thus, it would be of interest to use this method to evaluate strongyle community composition prior to and post anthelmintic treatment. 

This study aimed to use a fecal egg count reduction (FECR) test to assess the strongylid egg reappearance period (ERP) following ivermectin treatment within a large stud farm in Thailand. Additionally, the nemabiome metabarcoding technique adopted to study strongyle species infection patterns the after ivermectin treatment and highlight the specific species responsible for ‘early egg shedding’.

## 2. Materials and Methods

### 2.1. Ethical Approval and Informed Consent 

This study was conducted in Thailand between February and April 2023. The animal procedures used in this study were carried out in strict accordance with the ethical guidelines established by Chulalongkorn University in Thailand and were approved under the Institutional Animal Care and Use Committee (IACUC) protocol number 2331052. The head of the horse farm consented to partake in the study and provided details about the farm, the welfare of the animals, and the timing of the deworming protocol.

### 2.2. Study Design and Fecal Sample Collection

The study employed pre- and post-treatment fecal egg counts (FECs) to evaluate the ERP of strongyle parasites and monitor the differences in species composition before and after IVM administration. A cohort of 58 mixed breed horses, residing at the Horse Farm and Laboratory Animal Breeding Center of the Thai Red Cross Society in Phetchaburi, Thailand, was initially screened for this study. These horses comprised a diverse mix of males, females, and geldings aged between two and four years, and were housed in well-maintained grass pastures with year-round access to forage, water, and mineral salt blocks. Grain supplementation was only provided as needed to maintain their body condition. None of the subjects had received anthelmintics since August 2022. One week prior to the experiment, fresh fecal samples were collected rectally to determine the FECs for each individual horse (see Section 2.3). Horses with FEC values below 200 strongyle eggs per gram (EPG) are deemed to have a mild infection intensity and can be left untreated [47,48,49]. Therefore, only twenty-two horses with FECs ≥ 200 EPG were enrolled in the study. On day 0, these horses were randomly divided equally (n = 11) into two groups. In the first group, horses received IVM treatment at a dosage of 0.2 mg/kg (Nexmectin^®^ Oral Paste, AUDEVARD, Rue Médéric, France) adjusted to 115% of their body weight [31], to avoid underdosing. The second one served as the untreated control group for comparative analysis. For evaluation, the fecal samples were obtained on day 0, and subsequent sampling occurred at the weeks 2, 3, 4, 5, 6, 7, 8, and 9. The body weights of all animals were recorded both before and 9 weeks after treatment using a circumference tape (EquiVET^®^, KRUUSE, Langeskov, Denmark).

### 2.3. Fecal Egg Count, Larval Culture, and Harvesting

The collected horse fecal samples were transported in cooler boxes at (5–10 °C) to the parasitology unit at the Faculty of Veterinary Science, Chulalongkorn University, Bangkok, Thailand. The samples were refrigerated at 4 °C and analyzed within three working days. Strongylid fecal eggs were counted in duplicate using the McMaster technique [50], with a minimum detection limit of 50 EPG of feces. Four grams of feces were diluted in 56 mL of saturated salt solution with a density of 1.2. After homogenization, 1 mL aliquots were placed in McMaster chambers and examined under a light microscope (×10 magnification). The EPG count in the sample was determined by averaging the counts obtained from the duplicate procedures. 

To evaluate the differences of the strongyle species following IVM treatment, fecal samples were exposed to the larval culture in wide-mouth culture jars and incubated within a temperature range of 24–29 °C for seven days [51]. At each time point, the larvae harvested from each individual fecal sample were inspected under a stereomicroscope, washed, and then pooled (keeping the samples of the treatment and control group separated). The larvae were fixed in 100% molecular grade ethanol and stored at 4 °C in aliquots of ~2000 larvae for genomic DNA preparation.

### 2.4. DNA Preparation, ITS-2 Barcodes Amplification, and Sequencing

The genomic DNA was isolated from pooled larvae samples using the NucleoSpin^®^ DNA tissue kit following the manufacturer’s instructions (Macherey-Nagel, Düren, Germany). The purity and concentration of the DNA were assessed using a NanoDrop™ Lite spectrophotometer (Thermo Scientific, Waltham, MA, USA). The ITS-2 gene region was amplified via PCR using the NC1 and NC2 primers [52], a methodology previously outlined by [34,53]. A positive control used genomic DNA from morphologically identified adult parasites, while a negative control used water instead of a DNA template. Amplified PCR products were visualized by gel electrophoresis on a 1.5% agarose gel to ensure amplification. Then, PCR products were purified with NucleoSpin^®^ Gel and PCR clean-up according to the manufacturer’s protocol (Macherey-Nagel, Düren, Germany).

All libraries were combined at an equimolar concentration to ensure an equal number of raw reads for all samples. The Qubit Flex Fluorometer (Invitrogen, Waltham, MA, USA) and TapeStation 4200 (Agilent, Santa Clara, CA, USA) were used to verify the library product’s concentration and size. Sequencing was conducted on the Illumina MiSeq using a 500-cycle paired end reagent kit (MiSeq Reagent Kits v2) at a concentration of 15 nM with the addition of 15% PhiX Control. All protocols followed Illumina’s standard MiSeq operating protocol. The Fastq files comprising raw paired-end reads were generated and used for subsequent analysis.

### 2.5. Bioinformatic Analysis

ITS-2 amplicon sequencing data were analyzed using the free bioinformatics webserver, Galaxy Europe (https://galaxyproject.org, accessed on 15 October 2023) [54]. The inbuilt cutadapt tool [55] was used to remove the NC1 and NC2 primers from data sequences. The sequencing outputs were processed using the workflow of Divisive Amplicon Denoising Algorithm 2 (DADA2) to produce amplicon sequence variants (ASVs) [56]. In the first step, quality profile plots were checked to estimate quality filtering parameters. The filter and Trim tool was then used to eliminate reads with base quality scores of 2 or less, reads shorter than 200 base pairs, and readings with an estimated error rate of more than 2 in the forward direction or more than 4 in the reverse direction. The filtered reads were then used to train the system to recognize base-call errors within the dataset. This training involved utilizing the loess-error function, followed by the application of the core dada algorithm to correct these errors in the processed reads. The corrected forward and reverse reads were merged with a minimum overlap of 12 nucleotides and allowed a single error for each of the forward and reverse reads into an amplicon sequence variant (ASV) table. Chimeric sequences were removed based on consensus decisions, and the ITSx identification tool [57] was then used to remove regions flanking the ITS-2 gene. The sequence identification was performed using the assignTaxonomy function, enabling a 50% bootstrap cutoff, and referencing the public nematode ITS-2 database v1.5 (https://www.nemabiome.ca/its2-database.html, accessed on 22 March 2023). The nemabiome database was curated to resolve synonymous species names, eliminate incomplete sequences, and reduce the likelihood of errors as described by [44].

After data processing steps, ASV abundances were converted into proportional values. ASVs not assigned to any genus or species were removed from the dataset [53]. To ensure data quality, we only retained species with frequencies exceeding 0.05% for each sample. Since species abundance below 0.05% would relate to fewer than one L3 per sample, this was assumed to result from laboratory contamination or the misidentification of reads [58,59]. Subsequently, species proportions within each sample were recalculated by dividing the number of reads assigned to a species by the corrected total number of reads per sample.

### 2.6. Statistical Analysis

#### 2.6.1. Assessing IVM Effectiveness and Determination of ERP

The efficacy of IVM was evaluated two weeks post-treatment using the web-based tool at http://www.fecrt.com (accessed on 15 September 2023), as recommended by [31,60]. This tool incorporates the latest WAAVP guidelines, setting the expected efficacy for MLs in equine cyathostomins at 99.9% and a lower efficacy threshold of 96% [31].

The ERP was determined using mean percentage FECRs at different sampling times. The calculation was performed using a Bayesian hierarchical model through the webserver http://shiny.math.uzh.ch/user/furrer/shinyas/shiny-eggCounts/ accessed on 19 September 2023), as described by [61,62]. The interpretation followed the WAAVP guidelines [34], defining ERP as the week when the upper (CI) for the mean FECR dropped below the mean FECR recorded at two weeks post-treatment, reduced by 10% [34].

#### 2.6.2. Evaluation of Animal Weights

The body weights of the horses were estimated using a girth tape measurement at two time points: baseline (0 days) and 9 weeks after treatment duration. The mean body weights and standard deviations were calculated for the treatment and control group. A paired-samples *t*-test was used to evaluate the statistical significance of weight fluctuations pre- and post-treatment, assuming that the data were normally distributed. A significance threshold was declared if the *p*-value < 0.05. Boxplots were created using the “ggplot2” R package [63] to provide a visual insight into the spread of the data.

#### 2.6.3. Diversity Indices

Three statistical diversity indices (observed species richness, species evenness, and Shannon) were performed to describe differences in the strongylid community structure in terms of the treatment and control groups with regard to different time points. These indexes were calculated using the “vegan” R package [64] and excluding unassigned sequences [44]. Species richness (*S*) was defined as the total number of distinct species present in each sample. Evenness (*E*) measures the distribution of abundance among species. The formula for evenness involves dividing the Shannon’s index (*H*) by the natural logarithm of species richness (In(*S*)) [65]. Shannon’s diversity index is a composite measure of richness and evenness. The index was determined by utilizing the following formula: H=−∑isPiln ⁡Pi
where $P_{i}$ is the proportional abundance of species *i* [66]. The resulting index, *H*, ranges from 0 to a maximum value, with higher values indicating increased diversity. A value of 0 signifies a lack of diversity in the community [66].

## 3. Results

### 3.1. Fecal Egg Counts

A total of 58 mixed-breed horses were initially screened one week prior to deworming treatment. Among these, 22 horses had an FEC exceeding 200 EPG and were subsequently enrolled in this study (Appendix A). The FECRT data analysis from fecrt.com, using the Beta Negative Binomial (BNB) method (version B), classified the efficacy of the treatment as ‘Susceptible’ with no significant evidence of resistance (*p* = 0.197) but a clear susceptibility (*p* = 0.011) (Appendix A). The mean FECs for the treatment and control groups were assessed at each sampling time as presented in Figure 1. The mean percentage of the FECR values for the IVM treated group in each post-treatment time point is shown in Table 1. In this study, the reduction threshold for detecting the strongylid ERP was 89.4%, based on the FECR from two weeks after treatment (99.4%–10.0 = 89.4%). Notably, the treated group met the ERP criteria at six weeks after treatment, with the upper limit of the 95% CI (84.0) falling below the specified threshold. Nevertheless, it is worth highlighting that the first individuals that exceeded the set threshold were in the fourth week following the IVM administration. In addition, nearly fifty percent of the individuals revealed shortened ERPs at the fifth week post-treatment. At the last recorded time point, the treated group displayed an increase in the mean FEC surpassing the initial count observed on day 0 (see Appendix A and Figure 1).

In addition, the characteristic *Parascaris equorum* eggs were observed in two horses within the treatment group, with egg counts of 100 and 150 EPG before the IVM treatment. These eggs were not detected in subsequent FECs during the examination period post-treatment.

### 3.2. Animal Body Weights

Regarding changes in animal body weight throughout the study duration, the individual data for both the treatment and control groups are detailed in (Appendix A). Overall, prior to IVM administration, the mean body weight for the treatment group was 254.5 ± 59.04 kg, which marginally increased to 255.1 ± 55.81 kg post-treatment, yielding a non-significant *p*-value of 0.8. In contrast, the mean body weight of the control group was 250.6 ± 45.59 kg at day 0, which decreased to 246.8 ± 45.53 kg after the same observational period, demonstrating a significant reduction in body weight (*p* = 0.01) as depicted in Figure 2.

### 3.3. ITS-2 Nemabiome Metabarcoding and Diversity Results

Details concerning the number of sequenced reads/amplicons and number left at each step of DADA2 pipeline can be found in (Appendix A). In summary, between 359,401 and 156,915 pairs of reads were generated per sample, and between 113,530 and 54,117 pairs/amplicons were retained after bioinformatic analysis. These retained sequences were then clustered, resulting in a total of 601 ASVs. Among these ASVs, 515 were assigned to the genus level, and 484 were assigned to the species level using a 50% confidence threshold.

The nemabiome metabarcoding analysis revealed the presence of sixteen distinct species in the studied horse groups (control and treatment groups), as shown in (Appendix A and Figure 3). These identified species were *Coronocyclus* (*Cor.*) *coronatus*, *Cor. labiatus*, *Cor. labratus*, *Cyathostomum* (*Cya.*) *pateratum*, *Cylicocyclus* (*Cyc.*) *ashworthi*, *Cyc. brevicapsulatus*, *Cyc. insigne*, *Cyc. leptostomum*, *Cyc. nassatus*, *Cylicostephanus* (*Cys.*) *calicatus*, *Cys. goldi*, *Cys. longibursatus*, *Cys. minutus*, *Poteriostomum* (*Pot.*) *imparidentatum*, *Triodontophorus* (*Trio.*) *nipponicus*, and *Trio. serratus*. Notably, *Cyc. brevicapsulatus* was exclusively found only on the initial day (day 0) in both groups. All other 15 species were consistently found in the control group during the entire study period. In contrast, these species exhibited a distinct pattern in the treatment group. Specifically, *Pot. imparidentatum*, *Trio. nipponicus*, and *Trio. serratus* were no longer detectable following IVM treatment. At the fourth week post treatment, nine species were detected early, and among them, *Cys. longibursatus* and *Cyc. nassatus* constituted approximately 60% of the observed community. From the sixth week onwards, we observed the reappearance of *Cya. pateratum* and *Cyc. insigne* eggs, in addition to other Cyathostominae species. This pattern mirrored the composition of the pre-treatment dataset, with the exception of *Cor. labratus* species which was only detected after nine weeks and was absent among the species identified prior to the administration of the treatment.

The analysis of species diversity over time in both the treatment and control groups revealed distinct trends, as shown in Table 2. Species richness remained relatively stable in the control group. Conversely, in the treatment group, there was a dynamic pattern of change in the species richness over the time. Species evenness showed increasing values in the control group while the treatment group experienced a decline at 4 and 6 weeks followed by a modest recovery. Similarly, the Shannon diversity index reflected these trends, with the control group demonstrating an overall increase in diversity, while the treatment group experienced a temporary drop at 4 weeks followed by a gradual increase.

## 4. Discussion

The extensive use of anthelmintic drugs led to the emergence of anthelmintic resistance in parasitic nematodes worldwide [67]. Monitoring anthelmintic resistance in strongylid parasites remains a top priority in equine parasitology [68]. Nonetheless, there is a paucity of information regarding the IVM performance in treating strongyle infections in Thai horse populations. Therefore, this study aimed to determine the ERP of strongylid parasites and identify which species are involved using the nemabiome metabarcoding technique. This information is important to optimize the parasitic control measures and uncover the underlying molecular biology of anthelmintic resistance in this complex group of nematode parasites.

This study conducted an FECR test to assess the treatment effectiveness using the BNB method, which is able to analyze FECR data, even when there are less than three post-treatment observations above zero [69]. This method relies on separate statistical tests to calculate *p*-values for paired non-inferiority (susceptibility) and inferiority tests (resistance), using the predetermined expected efficacy and lower efficacy thresholds [21,54]. Our finding revealed that the tested population is susceptible to the treatment with a *p*-value of 0.01 for the susceptibility test, indicating the high efficacy of IVM against the targeted parasites. These findings are supported by the high FECs constantly detected in untreated control horses, thus ruling out any external influence on the parasitic burden occurred during the study period. Our study is consistent with prior research conducted on stud farms, where treatment efficacies of 99% or higher were reported in horses 14 days post-administration [18,42,68]. Nonetheless, the lower efficacy and emerging anthelmintic resistance of cyathostomins to IVM remain significant concerns in equine populations [70,71]. Especially, MLs remain the dominant class of drugs used in horses worldwide [72,73], which can lead to the extensive and continuous use of a single drug. This inappropriate practice is considered responsible for the rapid appearance of resistance [18].

Understanding the ERP for each anthelmintic drug is crucial for equine parasite management as treatments at intervals equal to or shorter than ERP can promote resistant strongyle populations [74]. This improper usage of drugs can lead to a reduction in parasite stages in refugia, which are not exposed to drugs at the time of treatment, although they are still susceptible. This refugia subpopulation lacks genes for resistance, potentially slowing the proliferation of resistant populations and extending the drug effectiveness by preserving a reservoir of drug-susceptible genes [74]. In light of prior reports, the assessment of ERP was calculated based on a predetermined fixed cutoff value of 90% for MLs [42,75], suggesting that anthelmintics should achieve a 100% FEC to be deemed effective, which ignores that low-shedder horses might take longer to reach the specified threshold or might never reach it. In this study, we conducted FECRTs on a weekly basis over a 9-week period and determined that the ERP was the week in which the upper 95% (CI) fell below the predetermined threshold (89.4%). This approach was selected in agreement with other authors [18,34] that it represents a reasonable and reliable approach, as this method is adapted to the effect of the administered drug on FECs of horses undergoing evaluation. Our study with IVM indicates a shorter ERP than that originally reported for an IVM-susceptible population of parasites (i.e., 8–13 weeks) [36,37]. In addition, the initial observed reduction in FEC exceeded the set threshold on the individual level during the fourth week following treatment in two horses and increased to encompass five horses by the fifth week after treatment. Nevertheless, the mean FECR for the whole population failed to reach the threshold of ERP at these time points. This finding aligns with other reports in which strongyle eggs were found in fecal samples 28 days after IVM-treatment [42,76,77]. These recent changes in drug performance may suggest the potential emergence of IVM-resistant strongyles in the studied population in the future. To mitigate this risk, it is essential to regularly monitor individual fecal samples for at least four weeks beyond the ERP of the last administered drug [73]. This practice is crucial for evaluating the potential egg shedding and contamination by each individual within the population. Additionally, taking measures to reduce the number of free-living larvae in the pasture, for instance, the regular removal of fecal matter every two weeks [42,78], may slow down the development of anthelmintic resistance within these parasite populations.

In the present study, we observed a slight increase in body weight in the treatment group after IVM administration, although this change did not reach statistical significance. This coincided with a temporary decrease in the mean FEC after treatment. Conversely, the control group experienced a significant decrease in body weight, which aligns with previous findings of weight loss in horses with high parasite burdens. However, this observation could be due to random events or other factors not accounted for in this study, rather than an effect of parasitism [79,80,81]. In addition, there were minor variations in age among the two examined groups, which could potentially impact the individuals’ body weights. Thus, further investigation is needed to understand the factors contributing to the differences in body weight.

The analysis of the nemabiome data yielded valuable insights into the differences in infection patterns of strongyle species following IVM treatment, shedding light on the decreased susceptibility of certain species. Firstly, the most notable observation was the absence of three species, namely *Pot. imparidentatum*, *Trio. nipponicus*, and *Trio. serratus*, following IVM-treatment, suggesting the higher susceptibility of these species to treatment. These findings can be supported by the continuous appearance of these species in the untreated control group. Secondly, 9 out of the 16 detected species were observed as early as four weeks, with *Cys. longibursatus* and *Cyc. nassatus* accounting for approximately two-thirds of the eggs detected post-treatment, highlighting the lower susceptibility of these species. This observation aligns with previous reports [42,76], which found that *Cys. longibursatus* and *Cyc. nassatus* were among the first species to return to egg shedding 28 days after IVM-treatment, explaining their primary role in the shortening of ERP. In contrast, *Cya. pateratum* and *Cyc. insigne* were not detected until six weeks post-treatment, despite their higher abundance before the treatment initiation. These findings contradict prior research, which has shown that Cyathostominae species with a relatively high prevalence before the administration of IVM tend to exhibit a shortened ERP of approximately 4–5 weeks following treatment [22,82]. Furthermore, *Cyc. brevicapsulatus* was identified only on the initial day of treatment in both groups, and *Cor. labratus* was only found in the last examined time point after treatment. It may be due to a unique temporal occurrence of these species. Hence, no definite conclusion can be made regarding the effect of IVM on these species.

The data obtained from the descriptive diversity analysis corroborates previous findings. Results from species richness, evenness, and Shannon indices were somewhat higher in the control group over time. Conversely, in the treatment group, these indices exhibited a declining pattern during the fourth- and sixth-weeks post-treatment, reflecting the effect of IVM on the structural community composition during these weeks. However, determining species-specific efficacy poses statistical challenges that necessitate further investigation, including the relative abundance of species among individual animals [31], whereas species identification was conducted in the current study at the group level.

## 5. Conclusions

This study provides evidence of the effectiveness of ivermectin in eliminating adult populations of strongyle parasites within two weeks post-treatment along with a significant reduction in ERP. Furthermore, nemabiome metabarcoding underscores the existence of varying degrees of susceptibility within the strongyle community following treatment. However, the genetic basis of IVM resistance remains inadequately resolved. Employing a more powerful approach, such as genome-wide or whole-genome sequencing, might be required to identify the genes associated with macrocyclic lactone resistance. These findings are crucial for shaping future strategies to address and curb the emergence and dissemination of anthelmintic resistance in Thailand.

## Figures and Tables

**Figure 1 animals-14-00574-f001:**
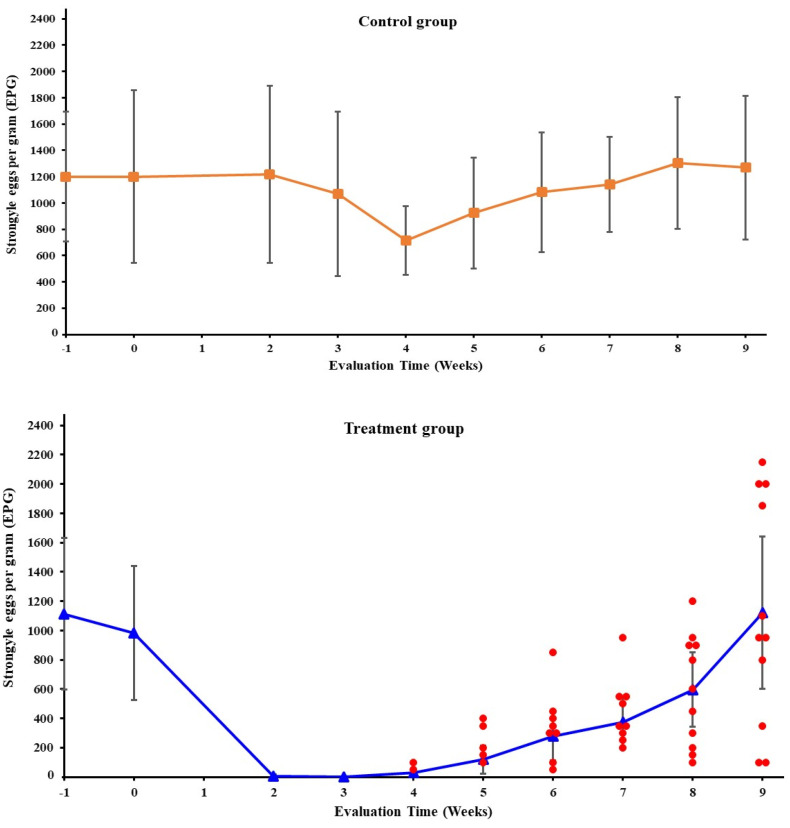
Dynamic changes in mean fecal egg counts of strongylid parasites throughout the study period. Two groups were observed: the untreated control group (squares) and the ivermectin-treated group (triangles). Treatment started in Week 0. Error bars designate 95% confidence intervals. The red dots represent horses exhibiting diminished susceptibility to ivermectin.

**Figure 2 animals-14-00574-f002:**
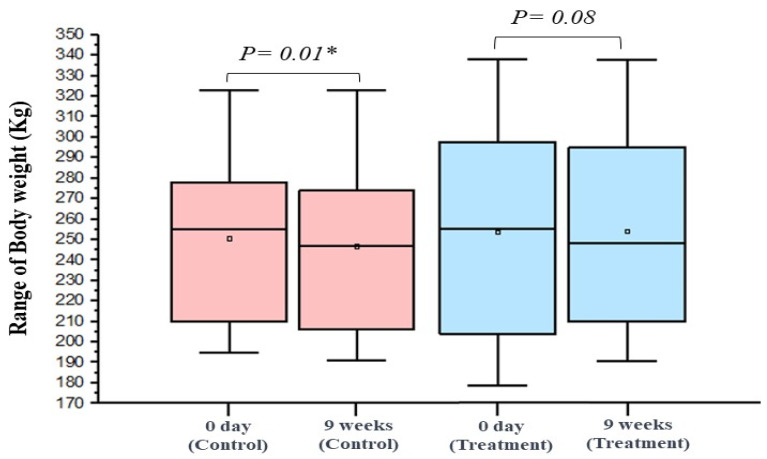
Boxplot depicting the distribution of horse body weights in the control and treatment groups at two time points—prior to the commencement of the experiment (0 day) and upon its conclusion (9 weeks). Significance indicated as follows: * *p* < 0.05.

**Figure 3 animals-14-00574-f003:**
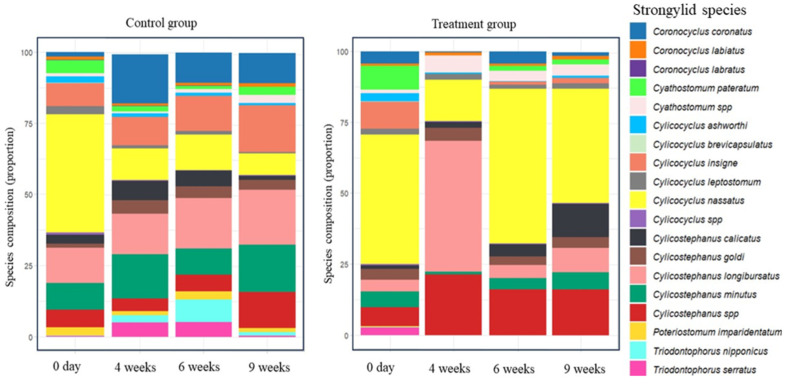
Stacked bar chart showing the relative abundance distribution of strongylid species in the untreated control group and the ivermectin treatment group at 0 day (pre-treatment) and 4, 6, and 9 weeks (post-treatment).

**Table 1 animals-14-00574-t001:** Dynamics of estimated mean fecal egg count reductions (FECRs) and associated 95% confidence intervals (CI) over a 9week period following ivermectin deworming in the treated group of horses.

Weeks Post-IVM-Treatment								
	2	3	4	5	6	7	8	9
FECR ^1^	99.4	99.8	97.1	87.7	70.4	57.1	37.9	26.2
CI ^2^	97.2–100	98.4–100	92.9–99.5	70.5–96.6	46.1–84.0	29.8–76.9	16.7–61.9	12.9–51.1

^1^ FECR, fecal egg count redaction. ^2^ CI, confidence interval.

**Table 2 animals-14-00574-t002:** Descriptive diversity indices for control and ivermectin treatment groups and at different time points.

Group	Control				Treatment			
Diversity Indices	0 Day	4 Weeks	6 Weeks	9 Weeks	0 Day	4 Weeks	6 Weeks	9 Weeks
Species richness	16	15	15	15	15	9	11	12
Species evenness	0.654	0.815	0.826	0.698	0.646	0.486	0.505	0.604
Shannon	1.813	2.207	2.239	1.891	1.750	1.068	1.211	1.501

## Data Availability

The Fastq sequencing data generated during this study are available in the GenBank/SRA database under the BioProject accession reference number PRJNA1012524 (https://www.ncbi.nlm.nih.gov/sra/PRJNA1012524 accessed on 28 December 2023).

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
