# Peer review of "Patterns of Equine Small Strongyle Species Infection after Ivermectin Intervention in Thailand: Egg Reappearance Period and Nemabiome Metabarcoding Approach"

_animals, 2024, doi:10.3390/ani14040574_

Round 1
Reviewer 1 Report
Comments and Suggestions for Authors
the considerations are in text

Reviewer 2 Report
Comments and Suggestions for Authors
Dear Authors,
Concerning your manuscript Animals-2821716 “Patterns of Equine Strongyle Species Reinfection After Ivermectin Intervention in Thailand: Egg Reappearance Period and Nemabiome Metabarcoding Approach” I believe it is an interesting topic and it provides new information on anthelmintic efficacy in domesticated horse in Thailand and about the species of intestinal strongyles more prevalent in these animals.
In my opinion the manuscript was well structured and I only have a few considerations
1. Introduction
Line 68: infection in current practice. Please add references.
Line 89: It would be more appropriate to define reduced ERP time rather than reduced efficacy.
Line 98: delate were. four cyathostomins… associated with
2. Materials and methods
2.2 Study design and Fecal sample collection
Line 132: to forage, water, pasture and. I suggest deleting pasture as it is a repetition of grass pastures.
2.3 Fecal egg count, larval culture and harvesting.
Line 155-155. Were the L3s also identified morphologically or were they just isolated?
3. Results
3.1. Fecal egg counts
In section 2.2 it was reported that 58 animals were evaluated while in this section 43 were evaluated. Please verify.
Reviewer 3 Report
Comments and Suggestions for Authors
Dear Editor, I have reviewed the manuscript “Patterns of Equine Strongyle Species Reinfection After Ivermectin Intervention in Thailand: Egg Reappearance Period and Nemabiome Metabarcoding Approach” submitted to be published on Animals. Here are my observations:
Title
Since the study only reports data relating to ciathostominae and the microscopic and molecular analyzes did not allow the authors to verify the presence of large strongyles, I recommend modifying the title appropriately
Materials and methods
add the formula to calculate the richness, evenness and Shannon's diversity index and the ranges of values of Shannon's diversity index and the references
Results
Figure 1- I propose to the authors to modify the figure by preparing two separate graphs, one for the control group and one for the treated group so that the horses with reduced sensitivity to ivermectin as identified by the red dots do not go on the graph relating to the control horses.
the group of treated horses consisted almost entirely (10 horses) of 2-year-old animals while the controls had only six 2-year-old horses. Weight changes could also be influenced by age and the presence of other parasites in these young subjects. Since the copromicroscopic examination used allows the diagnosis of other parasites in addition to strongyles, I advise the authors to add the overall results even if it does not fall within the scope of the study. Unfortunately, weight variations can be induced by multiple factors and in fact the authors should also add other information such as nutrition, time spent in the box or in the paddock which could help to understand these weight variations.
